# Profiles of Food Insecurity: Similarities and Differences across Selected CEE Countries

**Hanna Dudek** [1], **Joanna Myszkowska-Ryciak** [2,*] **and Agnieszka Wojewódzka-Wiewiórska** [3]

1   Department of Econometrics and Statistics, Institute of Economics and Finance, Warsaw University of Life Sciences (WULS), 02-776 Warsaw, Poland; hanna_dudek@sggw.edu.pl
2   Department of Dietetics, Institute of Human Nutrition Sciences, Warsaw University of Life Sciences (WULS), 02-776 Warsaw, Poland
3   Department of Development Policy and Marketing, Institute of Economics and Finance, Warsaw University of Life Sciences (WULS), 02-787 Warsaw, Poland; agnieszka_wojewodzka@sggw.edu.pl
*   Correspondence: joanna_myszkowska_ryciak@sggw.edu.pl; Tel.: +48-22-5937022

**Abstract:** Food security (FS) is influenced by primarily financial but also sociodemographic factors. Identification of correlates of food insecurity (FI) is a crucial issue in the context of achieving sustainable development goals. The aims of the study were: (1) to recognize FI in the selected Central and Eastern European (CEE) countries, (2) to examine common socioeconomic and demographic characteristics for FI. The analysis used the set of eight-item FI indicators adopted by the Food and Agriculture Organization, applying the Gallup World Poll survey data from 2017 to 2019. Multinomial logistic regressions were used to examine FI at mild and moderate or severe levels compared with FS. Differences in the profiles of FI were observed in analyzed countries: Poland, Lithuania and Slovakia. Lithuanians experienced the lowest FS, and Slovaks the highest. The FI status was associated with education, gender, age, household composition and income. It was found that the impact of these factors was not the same in the examined countries. Differences in profiles of FI in CEE countries indicate the need to analyze the problem individually for each country. Identifying groups particularly vulnerable to FI may allow appropriate targeting of instruments counteracting FI and adapt them to people with different characteristics.

**Keywords:** food insecurity; profiles; sustainable development goals; multinomial logistic regression models; relative-risk ratio; CEE countries; Poland; Lithuania; Slovakia; socioeconomic and demographic characteristics

## 1. Introduction

Ending global hunger and all forms of malnutrition is the most pressing global challenge [1]. Although current per capita global food production is estimated at 2796 kcal per individual per day, which is more than the minimum dietary energy requirement for adults [2], the global hunger rate is about one in ten people. Furthermore, since 2015, the global hunger rate has been on the rise. The problem is not only undernutrition caused by chronic insufficient supply of energy, but also malnutrition caused by insufficient supply of vitamins and minerals (micronutrient deficiencies) and overweight and obesity. Global estimates suggest that at current trajectories, all of these forms of malnutrition will increase globally from one in three persons in 2017 to one in two persons by 2030 [1]. It seems that it is not an insufficient global food production, but rather its inadequate distribution and utilization, that contributes to the problem of hunger and food insecurity. Thus, the need for urgent global remedial action has been included in the 2030 Agenda for Sustainable Development, adopted by all United Nations member states in 2015. The Sustainable Development Goal (SDG) No. 2 "Zero hunger" aims to end hunger and food insecurity, improve nutrition and promote sustainable agriculture by 2030 [3–5]. It is worth noting that hunger and food insecurity are related, but are not synonymous. Food security (FS) exists

when an individual has regular physical, social and economic access to sufficient, safe and nutritious food which meets his/her dietary requirements and food preferences for an active and healthy life [6]. On the contrary, food insecurity (FI) occurs when individual people and/or relatives in a household alter their food consumption or preferences because of a lack of physical or economic resources [7,8].

Food insecurity is a well-recognized cause of undernutrition and stunting [9]. While every human should be food secure, ensuring an adequate amount of nutritious food is especially important for women of childbearing age and children [3]. Insufficient intake of energy and/or nutrients, and consequently malnutrition, in these vulnerable groups has a large impact on the health condition of the entire population. Overall, malnutrition alters the immune system and increases vulnerability to infections in affected individuals at every stage of life [10]. In recent decades, FI has also been linked with obesity in high-income countries [11]. It is worth emphasizing that overweight and obesity as forms of malnutrition are recognized risk factors for severe course of COVID-19 at the individual level, also in young adults with no underlying conditions [12,13].

Although on a global scale, factors affecting FS include those controlling food production, i.e., climate change, the poor performance of the agriculture sector and poverty [6,14]; the FS status of an individual or a household is influenced by primarily financial but also sociodemographic factors and others (e.g., gender, time, employment skills, housing status, health condition, food skills or capabilities, health insurance status, social support, past economic hardship, aliment availability) [15–18]. However, these factors may have a different share in ensuring FS depending on the population/country/region.

### 1.1. Food Insecurity Assessment

Achieving the Sustainable Development Goals to a large extent depends on monitoring and follow-up processes [19–21]. Several markers and methods for assessing FI have been proposed so far to identify the problem and monitor progress in eradicating hunger and malnutrition, as well as to set goals for policy action at a national and international level. Although no generally accepted official measurement of FI in the world has been accepted [22], the Food Insecurity Experience Scale (FIES) was selected for monitoring Target 2.1 of the United Nations′ 2030 Agenda for Sustainable Development [3]. The FIES survey includes eight questions examining the individual respondent's experiences or the experiences of the respondent's household as a whole. These questions focus on the food-related behaviors and experiences reported by respondents and associated with increasing difficulties in accessing food due to resource constraints. The FIES is based on a well-grounded concept of the experience of FI composed of the following domains: 1. worry/anxiety; 2. changes in food quality; 3. changes in food quantity [23,24]. On the basis of the FIES scale, the risk of FI might be identified in individuals and communities in comparable way across different groups. Based on the FIES score (the number of positive answers to questions), the severity of FI can be access: ranging from zero (FS status) to eight (all symptoms of FI). The FIES score when analyzed in conjunction with the respondent and household characteristics can deepen the understanding of the risk factors and consequences of FI at an individual and household level [25,26].

### 1.2. Context for the Research

Nowadays, a growing body of literature on FI can be observed. There are a number of research papers on FI from a global perspective [25,27–31], as well as studies analyzing the situation in a group of countries [32–34] or a single country [26,35–39].

In the scientific literature, much attention has been allocated to FI in the less-developed countries of South Asia, Sub-Saharan Africa and Latin America [40–44]. Research on FI in developed countries has also begun to develop as the problem has been noticed and analyzed [22,45–49].

Research on FI in developing countries emphasizes the special role of agriculture in improving food availability and achieving food security [50]. The question arises

whether, in global terms, agriculture will keep up with the increased demand for food, reported with an increase in the population or migration [51]. At the same time, differences in the agricultural production potential that occur in individual countries are pointed out [50,52,53].

In the literature, various attempts in measuring FI can be found [31,45,54–56]. Thus, different indicators are used in FI studies across the world. Notwithstanding, regardless of indicators analyzed, most research shows that FI is strongly negatively associated with income [41,42,57]. However, some temporarily low-income individuals may be able to afford food by drawing down savings or incurring debts. The literature reveals that various individual assets have a significant relationship with FI [58,59]. Specifically, on the one hand, savings enhance the capacity for current consumption, but on the other hand, debts reduce it [59,60].

Most academics across the world indicate the important role of sociodemographic characteristics, such as location of dwelling, gender, age, high educational level and household composition as risk factors of FI [25,28,31,42]. Regarding location of dwelling, on one hand, in low-income economies, individuals living in rural areas seem to be more vulnerable than those living in cities and towns [31,42]. On the other hand, in upper middle-income economies, no significant difference between urban and rural areas regarding FI is found [31]. Analyzing FI in Europe, Grimaccia and Naccarato [61] found that FI in large cities was higher than in a rural location; however, the difference between those living in rural areas and small towns was not statistically significant. Similarly, results regarding gender are unambiguous—they depend on the examined population [25,28,31]. Specifically, Broussard [28] revealed higher mild FI among women that among men in the EU but no significant difference was found regarding moderate and severe FI.

Numerous scholars point out the inverted U-shape relationship between the age and the prevalence of FI, indicating less FI for younger and older individuals than for middle-aged [36,61]. However, this relationship was not confirmed in low-income economies by Smith et al. [31]. Furthermore, research regarding multiracial and multicultural countries revealed the effects of race, ethnicity and religion on FI [40,58,62].

Because some factors influencing FI may be unique to a certain country or region with specific socioeconomic, cultural and geographic settings, then there is a need to investigate the situation in various regions across the world.

### 1.3. The Picture of Analyzed Countries

The spatial scope of our research covers selected countries of Central and Eastern Europe (CEE): Lithuania (LT), Poland (PL) and Slovakia (SK) (Figure 1, Table 1). To the best of our knowledge, there is a lack of research on FI for these countries.

**Table 1.** Characteristics of analyzed countries: Lithuania (LT), Poland (PL) and Slovakia (SK).

| Indicator | LT | PL | SK |
|---|---|---|---|
| Population density * | 44.6 | 123.6 | 112.0 |
| Gini * | 35.4 | 28.5 | 22.8 |
| Real GDP growth rate (% average per year in 2014–2019) | 3.3 | 4.1 | 3.2 |
| GDP per capita in PPS * | 83 | 73 | 70 |

Source: own elaboration based on Eurostat, 2021 [63]; * data for year 2019.

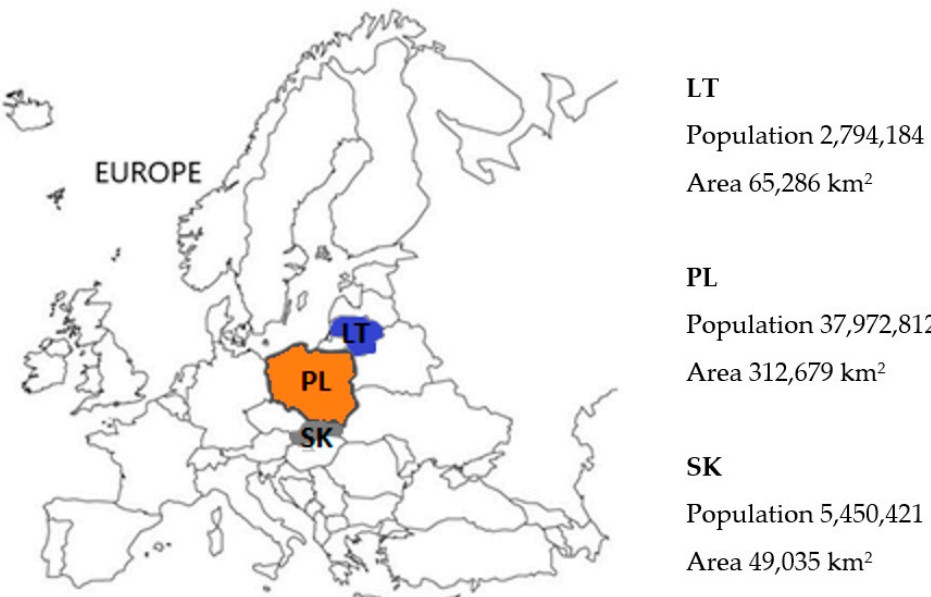

**Figure 1.** The study area and localization of analyzed CEE countries: Lithuania (LT), Poland (PL) and Slovakia (SK)—data for year 2019. Source: own elaboration based on Eurostat, 2021 [63].

Poland is the largest among the surveyed countries in terms of population; it also has the highest population density. The analyzed countries are located in close proximity: they constitute an area characterized by many cultural and political similarities related not only to the model of their socioeconomic development. Lithuania, Poland and Slovakia have common historical roots. Poland and Lithuania were part of the same country for many centuries; the same situation applies to Slovakia, which along with a large part of Poland belonged to the Austro-Hungarian Empire. Nowadays, these countries follow a similar path of economic transformation and integration with the European Union (EU), to which they have belonged since year 2004. They have managed to move from a centrally planned economy to a fully market economy, now with strong economic ties to major EU countries and the world economy. After the Second World War, Poland and Slovakia were satellites of the USSR, while Lithuania until 1990 was one of its republics. However, despite the indicated similarities, the CEE countries cannot be perceived as a fully homogeneous group [64]. There are clear differences in the level and dynamics of social and economic development, as well as internal differences in this respect.

According to the data for 2019, Lithuania showed one of the highest levels of income inequality among EU countries [63]. The Gini index of disposable income of households was 35.4, which makes Lithuania the second country (after Bulgaria) with the highest income inequalities in the EU. In turn, Slovakia is the country with the most even distribution of income in the European Union (with Gini coefficients at 22.8). In Poland, the indicator was 28.5, which is slightly less than the EU average.

The studied countries are also characterized by different dynamics of the economic development. In the years 2014–2019, the average annual GDP growth was the highest in Poland and amounted to 4.1%, whereas Slovakia and Lithuania had similar dynamics of development [63]. It is worth noting that Poland is the only country outside the eurozone; Slovakia has been in the eurozone since 2009, and Lithuania since 2015, respectively. To sum up, the presented data show that despite the similar geographic location and historical and cultural connections of all analyzed countries, there are differences in the shaping of various macroeconomic factors. This made us decide to analyze the issue of FI separately and create profiles for each country.

*1.4. The Purpose and Scope of the Study*

Our research is territorially oriented at selected CEE countries, i.e., Poland and its neighbors: Lithuania and Slovakia. The Czech Republic was excluded at the preliminary stage of the study due to the lack of availability of complete statistical data describing FS. Analyzed countries, as mentioned before, joined the EU in 2004, and are characterized by similar development conditions, but on the other hand, there are significant socioeconomic differences between them. The research area selected in this way allows to display the FI profiles. In the CEE countries, there is a lack of regular and methodologically consistent measurement of FI. Therefore, there is a need for investigations about the prevalence of FI and how it may be changing across countries. Combating FI and its associated consequences requires an understanding of the profiles of food-insecure individuals. Thus, our goal was to identify national profiles of food insecurity in countries under question.

The focus was on profiles of FI indicating whether people belong to the food secure category or the mild, moderate or severely food insecure categories. Multinomial regression models were estimated that aimed to explain the likelihood of belonging to each of the FI profiles.

The investigation of the socioeconomic and demographic factors affecting FI could help with the development of social policies to minimize the prevalence of FI in a given country. Thus, it is highly important to identify people being exposed to various forms of FI. In other words, vulnerable groups should be recognized.

We aim to answer the question: who are the food insecure in the CEE countries? Moreover, we want to examine which socioeconomic and demographic characteristics are common to the countries concerned, and which are not.

## 2. Materials and Methods

*2.1. Food Insecurity Measurement Methods*

The study uses Gallup World Pool (GWP) data for 2017–2019. Each country's sample size was about 1000 individuals, representative of the resident population aged 15 and older. More specifically, in 2014–2018 the sample size in each country was 1000, and in 2019—1080. The survey questions referring to various FI experiences are based on the Food Insecurity Experience Scale (FIES). The survey questions are presented in Table 2.

**Table 2.** Questions in the FIES.

| No. | During the Last 12 Months, Was There a Time When, because of Lack of Money or Other Resources: | Short Reference |
|---|---|---|
| (Q1) | You were worried you would not have enough food to eat | WORRIED |
| (Q2) | You were unable to eat healthy and nutritious food | HEALTHY |
| (Q3) | You ate only a few kinds of foods | FEWFOODS |
| (Q4) | You had to skip a meal | SKIPPED |
| (Q5) | You ate less than you thought you should | ATELESS |
| (Q6) | You ran out of food | RANOUT |
| (Q7) | You were hungry but did not eat | HUNGRY |
| (Q8) | You went without eating for a whole day | WHLDAY |

Source: FAO, 2021 [65].

The questions were asked to a nationally representative sample through face-to-face interviews. Respondents could answer either "Yes" or "No". FI experiences can be ranked in terms of severity from mild to severe (see Figure 2) [66]. In other words, the more food insecure a person is, the more likely she or he will report having suffered from the worst experience.

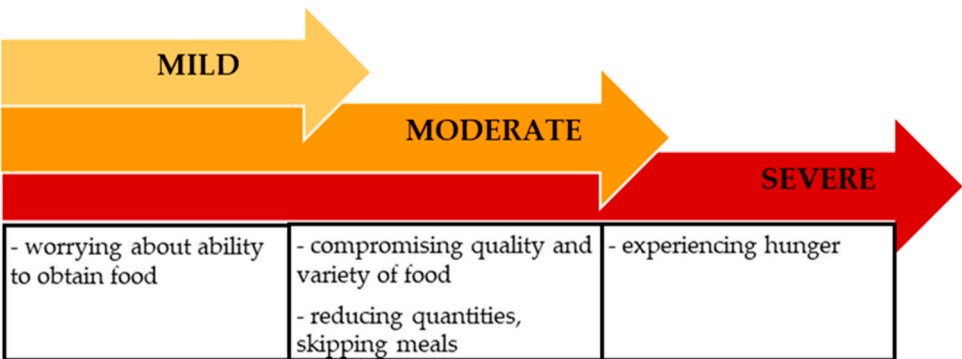

**Figure 2.** The range of severity of food insecurity. Source: [65,66].

The responses to the FIES questions are aggregated to produce raw scores ranging from 0 to 8. On this basis, FI is usually classified into 4 categories [67,68]:

1.   food secure with raw scores of 0;
2.   mild FI with raw scores of 1–3;
3.   moderate FI with raw scores of 4–6;
4.   severe FI with raw scores of 7–8.

However, as in the analyzed CEE countries the prevalence of severe FI is negligible (approx. 1% of the population), in our study we consider 3 categories:

1.   food secure with raw scores of 0 (FS);
2.   mild FI with raw scores of 1–3 (MFI);
3.   moderate or severe FI with raw scores of 4–8 (SFI).

Apart from FIES data, the Gallup World Poll database includes data relating to demographic and socioeconomic characteristics of individuals. Thus, the study examines the impact of various characteristics on FI.

The set of potential correlates includes: educational level, gender, age, location of dwelling, the income quintile group and household composition. Moreover, the influence of social capital is examined. This is done by using a binary variable which equals one if the individuals feel they can count on their family and/or friends in times of need.

The educational level was categorized as elementary (elementary or lower), secondary and tertiary (high or higher). The place of residence was classified as either city or suburbs, town, rural area or farm based on the current address of participants. The household composition included number of adults and number of children under 15.

*2.2. Methods*

In the first stage of statistical analysis, the associations between FI and categorical socioeconomic and demographic variables were assessed by using the chi-squared test. Moreover, Cramer's V measure was used to estimate the strength of these associations.

In the second stage of statistical analysis, multinomial logistic regressions were used to examine FI at mild and moderate or severe FI levels relative to food security. Despite that the analyzed dependent variable describing FI is ordered and could be investigated by ordinal logistic regression, we applied multinomial logistic regression instead. We chose this approach because:

1.   It does not impose the strong assumption of proportional odds (as ordinal logistic regression does);
2.   It provides the interpretation of results in terms of relative-risk ratios;
3.   Our goal is to investigate separate effects of socioeconomic and demographic characteristics on different levels of FI (i.e., mild and moderate or severe FI).

Multinomial logistic regression model assumes $m$ equations for the $m$ outcomes. Usually, it is assumed that the outcomes are coded using the set {1, 2,..., $m$} and that the

outcome of one is used as the base reference group. Thus, the probability of the response for the $i$th observation is equal to the $j$th outcome and can be expressed as [69]:

$$p_{ij} = P(y_i = j) = \left\{ \begin{array}{l} \frac{1}{1+\sum_{r=2}^{m} \exp(x_i \beta_r)} \text{ , if } j = 1 \\ \frac{\exp(x_i \beta_j)}{1+\sum_{r=2}^{m} \exp(x_i \beta_r)}, \text{ if } j > 1 \end{array} \right\} \tag{1}$$

where $x_i$ is the row vector of observed values of the independent variables for the ith observation, $i = 1, 2, \ldots, n,$

$n$ is the number of observations,

$\beta_j$ is the coefficient vector for outcome $j$, $j = 2, \ldots, m$.

The unknown parameters in each vector $\beta_j$ are typically jointly estimated using the maximum likelihood method [70,71]. However, their interpretation is hard because of nonlinearity in formula expressing probabilities in multinomial logit model. Therefore, the estimated coefficients are often transformed to relative-risk ratios (RRR), where the relative risk refers to the probability for each outcome of the dependent variable relative to the probability of the reference outcome ($j = 1$):

$$\frac{p_{ij}}{p_{i1}} = \frac{P(y_i = j)}{P(y_i = 1)} = \exp\left(x_i \beta_j\right) \tag{2}$$

The relative-risk ratio for multinomial logit can be defined as [69]:

$$RRR_{jk} = \frac{P(y_i = j | x_k + 1) / P(y_i = 1 | x_k + 1)}{P(y_i = j | x_k) / P(y_i = 1 | x_k)} = \exp\left(\beta_{jk}\right), j = 2, \ldots, m, \ k = 1, 2, \ldots, K, \tag{3}$$

where $\beta_{jk}$ is a parameter for the kth independent variable ($x_k$) corresponding to jth outcome (category),

$K$ is the number of independent variables included in model.

The RRR is independent of the particular values of covariates. A positive parameter $\beta_{jk}$ for an explanatory variable ($x_k$) implies an increased relative risk of observing an observation in category (outcome) $j$ rather than category 1 as $x_k$ rises by one unit, holding other covariates constant; a negative parameter $\beta_{jk}$ implies that the chance of being in the reference category (outcome) is higher relative to the jth category as $x_k$ increases by one unit.

In particular, if an independent variable ($x_k$) is a binary variable, then

$$RRR_{jk} = \frac{P(y_i = j | x_k = 1) / P(y_i = 1 | x_k = 1)}{P(y_i = j | x_k = 0) / P(y_i = 1 | x_k = 0)} = \exp\left(\beta_{jk}\right) \tag{4}$$

As noted in the last section, in our research we consider three categories ($m = 3$) and reference outcome is food security ($j = 1$). We conducted all statistical analyses using the STATA program (StataCorp LP, College Station, TX, USA). The GWP post-stratification national sampling weights have been applied to the survey data.

## 3. Results

A comparison between examined countries in terms of answers to individual FIES questions is presented in Figure 3. These eight questions (see Table 2) focus on respondents' behaviors and experiences related to the increasing difficulty in accessing food as a result of resource constraints. A positive answer to specific questions enables the classification of the FI intensity. The percentage of people who answered positively to all FIES questions was the highest in Lithuania. However, this difference was particularly evident in respect to questions No. Q1–Q3 and Q5. Comparing the situation of Poland and Slovakia, it can be noticed that in Poland, relatively more people than in Slovakia responded positively to questions No. Q1–Q3, while in Slovakia, there was a higher percentage of respondents answering positively to questions relating to moderate or severe FI (No. Q4–Q8).

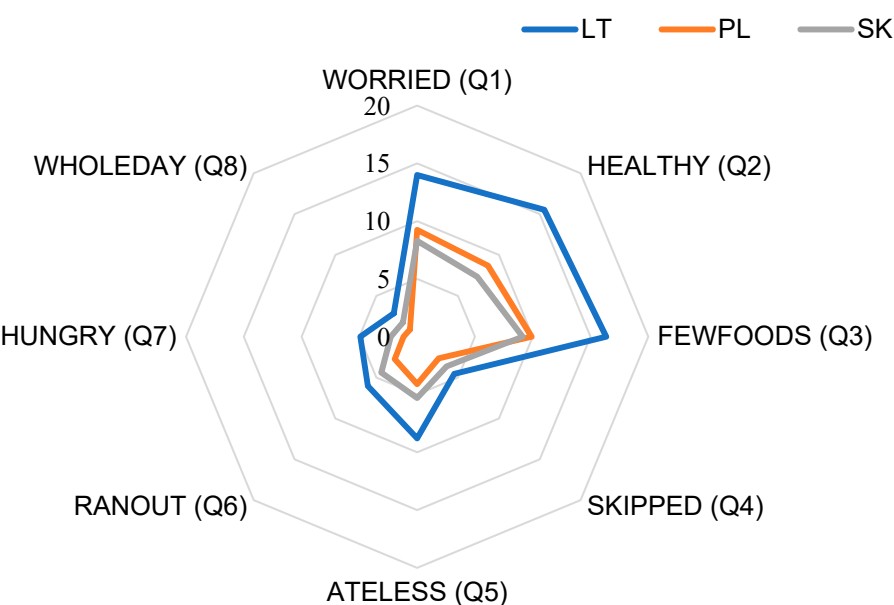

**Figure 3.** Percentage of positive responses to individual FIES questions (Table 2) in analyzed countries: Lithuania (LT), Poland (PL) and Slovakia (SK).

More condensed information on the prevalence of FS, mild FI and moderate or severe FI is presented in Figure 4.

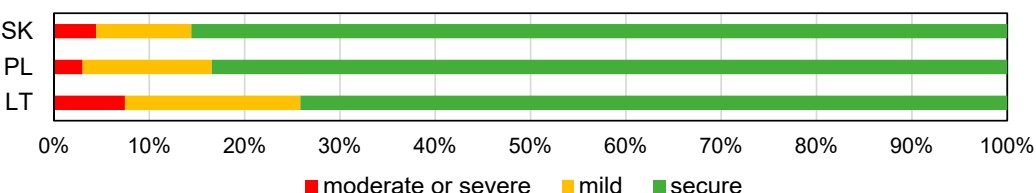

**Figure 4.** The prevalence of food insecurity in Lithuania (LT), Poland (PL) and Slovakia (SK) in 2017–2019.

Figure 4 shows that the lowest FS was experienced by Lithuanians, while the share of people with both mild and moderate or severe FI was the highest compared with other examined countries. The lowest percentage of people with moderate or severe FI was observed in Poland, while the lowest percentage of individuals with mild FI was identified in Slovakia.

More detailed information on the significance of differences between countries is presented in Figure 5. The prevalence of FS ranged between 72% in Lithuania and 84% in Slovakia. As all the confidence intervals for Lithuania do not overlap with ranges for Poland and Slovakia, this means that in Lithuania, the food security situation is different than in Poland and Slovakia (at all FI levels). On the other hand, when comparing Poland and Slovakia, no statistically significant differences were found in the proportions of food-secure and moderate or severe FI. Only for mild FI, the higher proportion for Poland compared to Slovakia was observed.

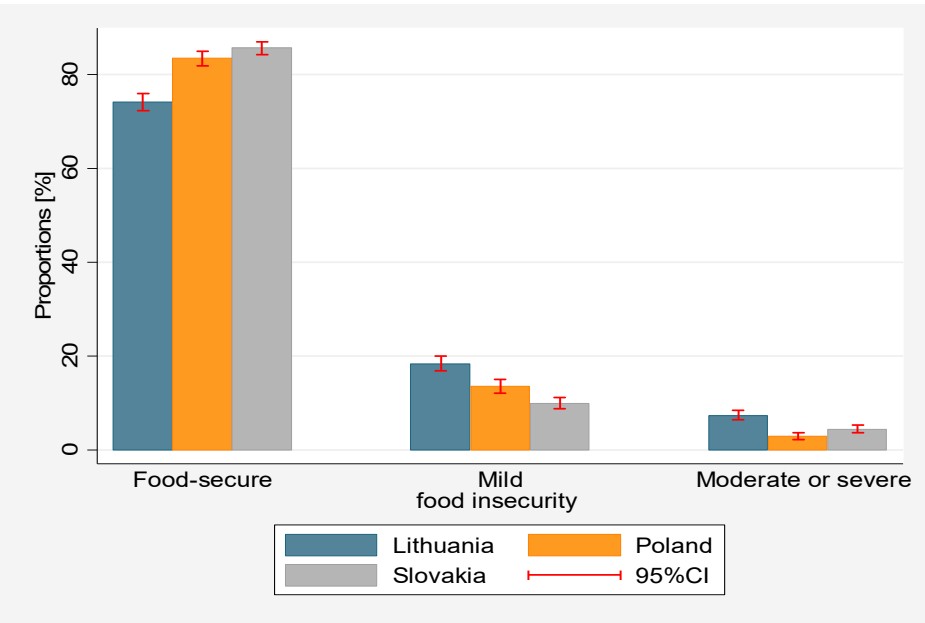

**Figure 5.** Confidence intervals associated with the specific FI category in 2017–2019.

Statistical analysis enables more detailed insight on the association between FI and individual characteristics. Therefore, in the subsequent stage, bivariate analyses were performed, including the chi-squared test (Table 3).

**Table 3.** The relationship between food insecurity and individual characteristics.

| Variables | Lithuania | | Poland | | Slovakia | |
|---|---|---|---|---|---|---|
| | χ2 Statistics * | Cramer's V | χ2 Statistics * | Cramer's V | χ2 Statistics * | Cramer's V |
| Gender | **9.35** | 0.06 | **6.68** | 0.05 | **7.16** | 0.05 |
| Number of adults in household | **136.77** | 0.15 | **72.75** | 0.11 | **73.32** | 0.11 |
| Number of children in household | **80.78** | 0.11 | **44.93** | 0.09 | **59.16** | 0.10 |
| Education | **52.63** | 0.09 | **79.44** | 0.11 | **56.28** | 0.10 |
| Location of dwelling | **33.18** | 0.07 | 5.67 | 0.03 | **13.58** | 0.05 |
| Social capital | **69.38** | 0.15 | **45.90** | 0.12 | **145.35** | 0.22 |
| Income quintile | **316.42** | 0.23 | **157.29** | 0.16 | **265.49** | 0.21 |
| Age | **70.88** | 0.11 | **42.83** | 0.08 | **28.12** | 0.07 |
| Years | 2.80 | 0.02 | **68.35** | 0.11 | **13.84** | 0.05 |

* Note: numbers marked in bold indicate results that are significant at a 0.05 level.

Significant differences were noted between FI and the majority of the examined variables, namely: gender, household composition, education, social capital, income and age. Nonetheless, for place of residence, no significant difference was noted in Poland. Cramer's V measure revealed that among categorical variables, social capital and income showed the strongest association with FI. Contrarily, the weakest relationship referred to gender and location of dwelling.

In the next stage of our study, the multinomial logistic regression model was used to examine differences in individual characteristics in countries under question. Results of multinomial logistic regression analyses are presented in Table 4. Two comparisons are shown between FI categories. The first set of coefficients predict mild FI (MFI) versus FS, the second set of coefficients compares moderate or severe food insecurity (SFI) to food security. The description of the results focuses on comparisons between mild food insecurity and food-secure individuals and between moderate or severe food insecurity and food-secure individuals. These comparisons examine whether: (1) mildly food-insecure individuals are

different to food-secure individuals; (2) moderate or severely food-insecure individuals are different to food-secure individuals. Table 4 summarizes the results of the multinomial logistic regression of individuals' socioeconomic and demographic characteristics and shows the relative risk ratios (RRR).

**Table 4.** Relative risk ratios from multinomial logistic regression.

| Variable | Mild FI vs. FS | | | | | | Moderate or Severe FI vs. FS | | | | | |
| | LT | | PL | | SK | | LT | | PL | | SK | |
| | RRR | SE | RRR | SE | RRR | SE | RRR | SE | RRR | SE | RRR | SE |
|---|---|---|---|---|---|---|---|---|---|---|---|---|
| Women | **1.26** | 0.15 | **1.44** | 0.19 | 1.25 | 0.16 | 1.30 | 0.24 | 1.50 | 0.42 | 1.06 | 0.21 |
| Number of adults in household | **0.72** | 0.04 | 0.90 | 0.07 | **0.76** | 0.05 | **0.72** | 0.08 | **0.62** | 0.10 | **0.65** | 0.07 |
| Number of children in household | **1.28** | 0.09 | **0.82** | 0.08 | **0.79** | 0.08 | **1.40** | 0.12 | 0.73 | 0.14 | 0.91 | 0.10 |
| Social capital | 0.79 | 0.13 | **0.50** | 0.08 | **0.40** | 0.07 | **0.35** | 0.07 | **0.32** | 0.09 | **0.17** | 0.04 |
| Age (ref. Age below 35) | | | | | | | | | | | | |
| Age 35–44 | 1.40 | 0.25 | 1.10 | 0.23 | **1.63** | 0.34 | 1.53 | 0.42 | 1.19 | 0.56 | 1.08 | 0.36 |
| Age 45–54 | 1.30 | 0.25 | 1.38 | 0.29 | 1.63 | 0.29 | **1.95** | 0.54 | 1.44 | 0.64 | 1.46 | 0.49 |
| Age 55–64 | **1.67** | 0.33 | **1.69** | 0.35 | **1.52** | 0.33 | 1.49 | 0.47 | 1.17 | 0.55 | 1.33 | 0.44 |
| Age 65–74 | 1.21 | 0.25 | 1.53 | 0.37 | 1.33 | 0.30 | **2.08** | 0.65 | 0.97 | 0.46 | 1.55 | 0.52 |
| Age 75 + | **1.69** | 0.39 | 0.92 | 0.37 | 1.16 | 0.33 | **2.79** | 0.94 | 0.60 | 0.40 | 1.41 | 0.56 |
| Education (ref. Secondary) | | | | | | | | | | | | |
| Tertiary | **0.59** | 0.10 | **0.53** | 0.10 | 0.92 | 0.18 | **0.57** | 0.15 | **0.26** | 0.14 | **0.41** | 0.18 |
| Elementary | 1.47 | 0.32 | 1.21 | 0.28 | 1.11 | 0.21 | 1.63 | 0.51 | **2.29** | 0.84 | **2.12** | 0.53 |
| Location of dwelling (ref. Cities or suburbs) | | | | | | | | | | | | |
| Towns | **0.52** | 0.07 | 1.02 | 0.15 | 0.97 | 0.14 | **0.34** | 0.07 | 0.63 | 0.18 | **0.51** | 0.11 |
| Rural areas | 0.72 | 0.13 | 0.82 | 0.17 | 1.06 | 0.19 | **0.44** | 0.13 | **0.29** | 0.16 | 0.67 | 0.18 |
| Income quintile group (ref. Fifth quintile group) | | | | | | | | | | | | |
| First quintile group | **4.90** | 1.06 | **5.71** | 1.46 | **10.05** | 2.50 | **13.20** | 5.03 | **51.94** | 27.66 | **20.71** | 8.07 |
| Second quintile group | **4.50** | 0.91 | **2.74** | 0.68 | **4.06** | 0.96 | **4.31** | 1.65 | **7.90** | 4.64 | **4.30** | 1.71 |
| Third quintile group | **2.38** | 0.50 | **2.89** | 0.67 | **3.08** | 0.72 | **2.72** | 1.12 | **4.40** | 2.69 | 1.64 | 0.70 |
| Fourth quintile group | **1.59** | 0.32 | **2.38** | 0.55 | 1.37 | 0.35 | 1.98 | 0.81 | **7.41** | 4.21 | 1.40 | 0.62 |
| Years (ref. 2017) | | | | | | | | | | | | |
| 2018 | 1.05 | 0.16 | **0.29** | 0.05 | **1.66** | 0.26 | 1.29 | 0.31 | **0.46** | 0.15 | **1.91** | 0.48 |
| 2019 | 0.87 | 0.13 | **0.38** | 0.06 | **1.66** | 0.26 | 0.69 | 0.17 | **0.20** | 0.08 | **2.19** | 0.54 |
| Constant | **0.24** | 0.07 | **0.21** | 0.07 | **0.09** | 0.03 | **0.11** | 0.05 | **0.06** | 0.05 | **0.10** | 0.05 |

Note: RRR are relative risk ratios, SE-standard errors, numbers marked in bold indicate results that are significant at a 0.05 level.

The results shown in Table 4 reveal interesting findings. Firstly, when comparing the results for mild FI regression and moderate or severe FI regression, it can be noticed that some socioeconomic and demographic characteristics significantly influenced one process but not another. For example, in the model for LT and PL, gender was significant in mild FI regression, but at the same time was insignificant in moderate or severe FI regression. Secondly, the set of significant correlates was not the same in all countries under question. In particular, location of dwelling was rather significant in the model for LT, but this finding was not confirmed for PL and SK, especially in mild FI regression.

A detailed interpretation of the impact of various characteristics can be made on the basis of an RRR computed using formula 3. The RRR is interpreted as the effect of a one-unit change in the explanatory variable on the probability of being in the dependent variable outcome (category) under consideration, compared with the reference outcome (category). In our study, on the one hand, an RRR that is less than one indicates that there is a lower likelihood of being insecure than the likelihood of being food secure. On the other hand, an RRR greater than one means that there is a greater likelihood of being insecure

than being food secure. For example, considering the gender effect for Lithuania, we found that (see formulas 3–4):

$$RRR_{2. \ Women} = \frac{P(y_i = 2|Women = 1)/P(y_i = 1|Women = 1)}{P(y_i = 2|Women = 0)/P(y_i = 1|Women = 0)} = 1.26 \ \text{or equivalently}$$

$$\frac{P(y_i = 2|Women = 1)}{P(y_i = 1|Women = 1)} = 1.26 \cdot \frac{P(y_i = 2|Women = 0)}{P(y_i = 1|Women = 0)}$$

This result means that the relative risk of being mildly FI comparing with being food secure in Lithuania was 1.26 times (i.e., 26%) greater among women than among men when holding other predictors constant. The considered relationship can be also expressed as:

$$\frac{P(y_i = 2|Women = 1)}{P(y_i = 2|Women = 1)} = 1.26 \cdot \frac{P(y_i = 1|Women = 0)}{P(y_i = 1|Women = 0)}$$

What can be interpreted is that Lithuanian women compared with Lithuanian men were 26% more likely to be mildly food insecure than food secure.

Thus, on the basis of the model, we may state that the relative risk of being mildly FI rather than being food secure was greater among women that among men: in Lithuania by 26% in Poland by 44%; however, in Slovakia there was no significant difference between gender. This relative risk decreased if the number of adults in household increased by one person: in Lithuania by 28%, in Slovakia by 24%; however, in Poland there was no statistically significant relationship in this regard. At the same time, the mild FI relative risk increased by 28% in Lithuania if the number of children in the household increased by one child, but decreased in Poland and Slovakia by 18% and 21%, respectively. For those individuals that felt they could count on their friends and family in times of need, the relative risk was twice as low in Poland and 60% lower in Slovakia. By contrast, social capital in Lithuania was not a statistically significant factor. Regarding the age, the relative risk was greater for older persons than for those aged below 35; specifically, in all countries for individuals aged 55–64, the risk was greater at about 60%. In the case of education, the relative risk of mild FI in Lithuania and Poland was about twice as low among individuals with tertiary education compared with secondary education; nevertheless, in all three countries there was no statistical difference between those with secondary and elementary education. In Lithuania, the relative risk was about twice as low among individuals living in towns compared with people living in cities or suburbs; however, in Poland and in Slovakia there was no statistically significant relationship in this regard. As expected, the RRR was higher for individuals with lower income; specifically, comparing with the fifth quintile group, the RRR for people from the first quintile group was about five times higher in Lithuania and in Poland and tenfold higher in Slovakia. To sum up, the relative mild FI risk was subject to a downward trend in Poland and growing trend in Slovakia, while none of the significant trend was observed in Lithuania during the 2017–2019 period. Thus, when analyzing mild FI, it was found that, in principle, in relation to each of the examined factors (gender, household composition, social capital, age, education, location of dwelling, income quintile group), there are differences in the analyzed countries.

A similar interpretation for the RRR corresponding to the next FI category was performed. We may notice that the relative risk of being moderately or severely FI rather than being food secure did not differ in terms of gender in all three countries, but decreased if the number of adults in household increased by one person: by 28% in Lithuania, 38% in Poland and 35% in Slovakia. Corresponding RRR increased by 40% in Lithuania if the number of children in the household increased by one child, while in Poland and Slovakia there was a lack of significant relationship in this regard. Additionally, the relative risk was about three times smaller in Lithuania and Poland and more than five times smaller in Slovakia for those individuals that felt they could count on their friends and family in times of need. In the case of age, the RRR was more than twice as high among individuals over the age of 65 compared with individuals aged below 35 in Lithuania; nevertheless, in Poland and in Slovakia there was no significant relationship in this regard. Regarding education, the RRR

was lower among individuals with tertiary education compared with secondary education (RRRs: Poland 0.26, Lithuania 0.57 and Slovakia 0.41), and it was about twice as high among individuals with tertiary education among Poles and Slovaks, while in Lithuania, there was no significant relationship between elementary and secondary education. The RRR was about threefold lower among individuals living in towns and 56% lower among villagers compared with people living in cities or suburbs in Lithuania; in Poland, it was 71% lower among villagers, and in Slovakia about twice as low among individuals living in towns compared with people living in cities or suburbs. In all countries, the RRR was higher for individuals with lower income. Summing up, relative risk of being moderately or severely FI showed a downward trend in Poland during 2017–2019, and a growing trend in Slovakia; however, none of this significant trend was observed in Lithuania during the analyzed period.

Based on the above results, it can be concluded that when it comes to moderate or severe FI compared with mild FI, there were more common statistically significant factors for all three countries. Specifically, in all countries, a relevant role of social capital, higher number of adults in household, higher education comparing with secondary education, rising income in decreasing the likelihood of moderate or severe FI were noted. Moreover, no gender differences in all three countries, ceteris paribus, were found.

Based on the results of multinomial logistic regressions estimates, we have designated four exemplary types of individuals (Table 5). Obviously, these sample types do not exhaust all possible values of the explanatory variables included in the models. However, they provide insight into how different FI profiles are for people with different demographic, social and economic characteristics.

**Table 5.** Types of individuals based on demographic, social and economic characteristics in the year 2019.

| Characteristics | Type 1 | Type 2 | Type 3 | Type 4 |
|---|---|---|---|---|
| Gender | Man | Woman | Woman | Man |
| Number of adults in household | 1 | 1 | 3 | 3 |
| Number of children in household | 0 | 0 | 3 | 2 |
| Social capital | bad | bad | good | good |
| Age | 55–64 | Above 75 | 45–54 | Below 35 |
| Education | Elementary | Elementary | Tertiary | Tertiary |
| Location of dwelling | Cities or suburbs | Cities or suburbs | Towns | Towns |
| Income quintile group | First | First | Third | Fifth |

The first two types refer to individuals with the worst traits, the third to average and the fourth to rather favorable. It should be noted, however, that the favorable discriminants are not always the same in the analyzed countries, e.g., the number of children. The common unfavorable traits included: no children in the household, one-person household, elementary education, living in cities or suburbs and low income. The determinants predisposing to FS included: presence of three adults and more than one child in the household, tertiary education, living in a town, a higher level of income and age under 54 years.

For these four distinguished types, according to formula (1), we determined the probabilities corresponding to different FI profiles. Based on the estimated probabilities, we developed percentages charts (i.e., probabilities expressed in percentages) (Figure 6).

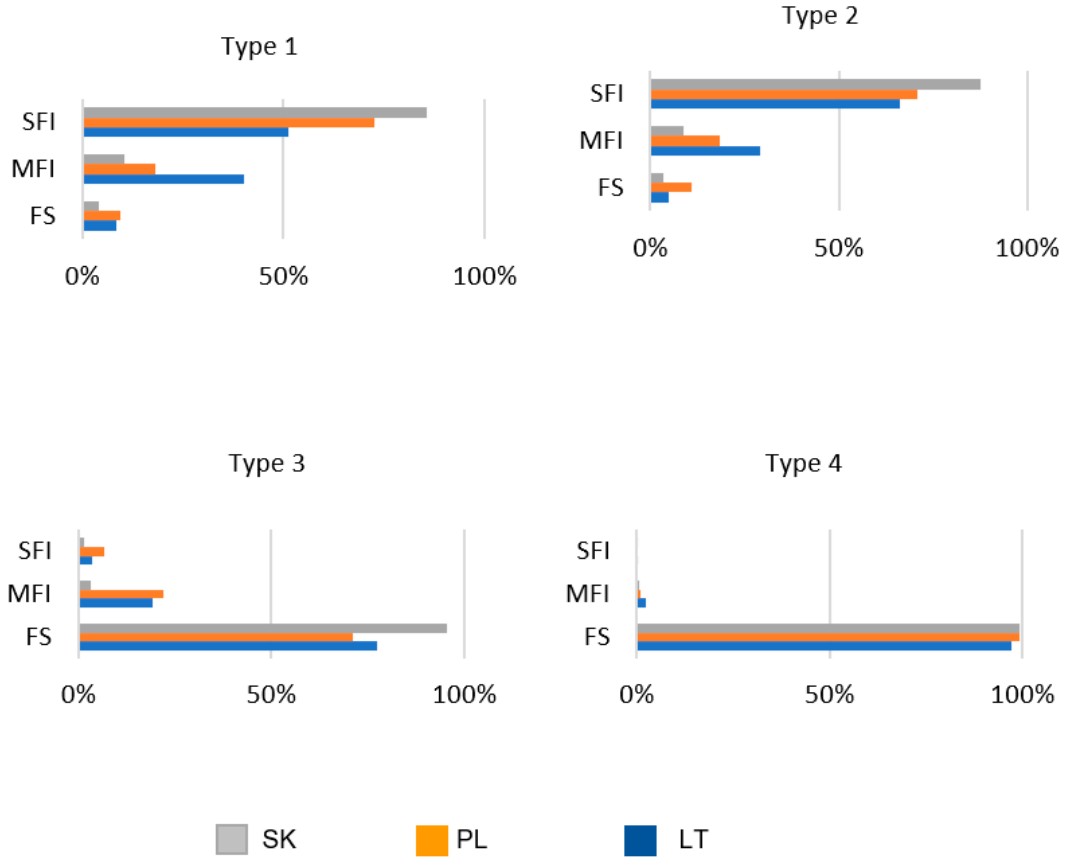

**Figure 6.** Predicted percentages of individuals in analyzed CEE countries: Slovakia (SK), Poland (PL,) Lithuania (LT) and according to specified FI profiles (SFI—moderate or severe food insecurity; MFI—mild food insecurity; FS—food security).

The predicted probability of experiencing a specific FS situation for all distinguished types, except type 4, varies from country to country, with the highest differences for type 1. For this type, the predicted probability of experiencing moderate or serious FI was the highest for Slovakia and the lowest for Lithuania. On the contrary, the predicted probability of experiencing mild FI in Lithuania was more than 2–2.5 times higher compared with Poland and Slovakia, respectively. On the other hand, the predicted probability of FS for type 1 was approximately two times lower for Slovakia compared with Poland and Lithuania. Similarly to type 1, type 2 also indicated a set of discriminants negatively influencing the FS situation. In this type, we estimated a similar probability pattern of experiencing different FS situations, but the differences between countries were not so marked. The second difference is that Poles with the characteristics defined as type 2 are about twice as likely to experience FS as compared with Lithuanians and Slovaks. For type 3, the probability of FS is the highest for Slovakia, where the predicted probability of moderate and severe FI was extremely rare in this country. For type 4, the probability of moderate and severe FI or mild FI is extremely low ($\leq$1%). Only in the case of Lithuania, 3% of individuals belonging to type 4 can be projected to be mildly food insecure.

## 4. Discussion

Based on the obtained results, it can be stated that there are differences in the profiles of FI in analyzed CEE countries. The Lithuanians experienced the lowest FS, and the Slovaks the highest. The differences found can be explained to some extent by the differences in the living conditions of the population in the analyzed countries. Lithuania clearly stood out among the countries studied: in the case of many characteristics, the presence of FI was more common in this country. Explaining the worse situation of Lithuania compared with Poland and Slovakia, one should mention differences in the level of material deprivation,

on the basis of which Lithuania is classified as one of the countries with a high level, while Poland and Slovakia are among the countries with an average level [63,72]. This can significantly determine FI and explain the identified differences in terms of the FI profile. It is worth adding that the prevalence of FI identified in the surveyed countries corresponds to the variation in the percentage of people at risk of poverty or social exclusion, which in 2018 was by far the highest in Lithuania (28.3%), and lower in Poland (18.9%) and in Slovakia (16.3%). This proves the relationship of FI with the economic situation of the population and shows how complex the conditions for the presence of FI can be. These results are also confirmed by the subjective assessment of life satisfaction, which in Lithuania is described as low by as many as 36% of the population [63]. The average rating of life satisfaction in this country is also relatively low (6.4 points), which is the lowest among the countries surveyed and low compared with the EU average (7.3). Lithuania is also characterized by the lowest percentage of public social spending in GDP [73] and still (despite an increase in recent years) the lowest share of social benefits to households as a percentage of GDP, which may be important for the prevalence of FI.

The reasons for the identified differences between the analyzed countries in the prevalence of individual FI forms may also result from different situation of individuals. The explanation can be seen in the existence of large disparities in disposable income in the surveyed countries or differences in food prices (e.g., the greatest income stratification and disadvantageous income situation of pensioners in Lithuania [74,75], and the best situation in Slovakia, which is the country with the most even income distribution in the European Union. The individual situation with regard to the burdening of household income with fixed charges (e.g., loan repayments, fixed house maintenance fees) may be important, especially for the existence of moderate or severe food security. In the case of mild food security, its prevalence may depend on the type of typically consumed and important food products (e.g., considered as healthy, with a high nutritional value), their availability and prices on domestic markets, which may significantly differ in the analyzed countries due to, e.g., traditions, culture or religion.

In our study, we focused primarily on the analysis of the impact of individual socioeconomic and demographic characteristics on the presence of FI. It turned out that specifically, compared with men, women in Lithuania and Poland have a greater propensity to be classified as mildly food insecure rather than food secure. Our findings are in line with those of Broussard [28], who revealed the gender difference in terms of mild FI in the EU but no significant difference regarding moderate and severe FI. A potential explanation for this may include the specific role of women in the household, who are more often than men responsible for housekeeping and day-to-day food supply decisions. Only the exceptionally difficult situation of the family in terms of FI (moderate or severe) engages all household members, regardless of gender.

Contrary to our expectations, the same importance of children in households in the context of the prevalence of FI was not found in the analyzed countries. It turned out that outside Lithuania, the number of children is not significant for moderate or severe FI. Moreover, in Lithuania, FI, regardless of its severity, increased if the number of children in the household increased. The example of this country shows how important it is to ensure that FS is the national (social) policy in the field of supporting families with children. Lithuania did not have a universal child benefit system until 2018, and the existing tax instruments were, in practice, difficult for parents to use [76]. This was accompanied by a low level of enrollment in the case of young children [77], which could limit the economic activity of parents on the labor market, especially single parents, who are also at the highest risk of poverty in Lithuania [78]. Due to the introduction of a child support system in Lithuania in 2018 ("child money"), an improvement in the income situation of families with children can be expected in the future, which may affect the occurrence of FI.

Our other findings are largely consistent with empirical studies that use GWP FIES data [28,31]. The results of our study revealed that generally FI was greater for older persons than for those aged below 35. As an explanation, it can be indicated that the

income of older people grows slower than in other age groups. Additionally, pensions are usually lower than wages and, as shown by the example of Lithuania, pensions grow more slowly as the average wage increases [75,79]. This can make the situation of older people more difficult, as well as their subjective feelings worse. This is also confirmed by the large percentage of households of people over 65 in all surveyed countries (compared with total households) that have difficulties making ends meet [63].

Similar to Miller et al. [80] and Smith et al. [31], our study showed that midlife appears to be a period of increased vulnerability to FI. Specifically, in all countries, for individuals aged 55–64, the relative risk of being mildly FI rather than being food secure was greater at about 60%. The age range of 55–64 is classified as late middle age, in which, as evidence shows, adults experience many changes in their lives (in the social, psychological and biological spheres), which may increase the risk of FI [80]. At this age, the number of social roles increases [80,81]; this is called the sandwich generation, where people take care of both aging parents and children, and often also grandchildren, while combining it with professional work.

In line with Smith et al. [31], we found that the largest increase in the likelihood of experiencing FI was associated with low income, low social capital and low levels of education. This may be due to the fact that better educated people are more aware of the importance of their lifestyle, especially nutrition, for health and well-being. People having access to social support when their resources are constrained experience lower FI. As nutritional needs are elementary, belonging to any of those groups results in help for vulnerable individuals, which reduces the risk of experiencing FI.

An interesting issue in our research is the lack of dependence, apart from Lithuania, between mild FI and the place of residence, which has already been indicated in other studies [36]. In the case of moderate and severe FI, the risk of its occurrence is higher in large cities and suburbs, which is confirmed by the study of Grimaccia and Naccarato [25]. This may be associated with higher costs of living in large cities than in towns or rural areas. Moreover, it should be emphasized that the studied countries are characterized by a different territorial division and a different distribution of urban and rural population, which may explain the obtained results. Summing up, it can be stated that the analyzed socioeconomic and demographic characteristics determined a different degree being exposed to various forms of FI in the studied countries.

When analyzing the FI profiles, it would be worthwhile to take into account the impact of other characteristics, e.g., financial burden on respondents (loans, expenditure on some goods, for example, those which are health-related). Apart from the information on the number of adults and children, it is also worth taking into account the biological type of the household in which the respondent lives (i.e., whether the household consists of parents and children, or if it is a multigeneration unit, etc.). Unfortunately, the GWP lacks detailed information on the above-listed issues.

In conclusion, it should be noted that the obtained results in terms of FI are important in the context of the COVID-19 pandemic. Preliminary research in such countries as Brazil [82], Mexico [67], Ethiopia [83] and the United States [84] revealed that COVID-19 and national lockdowns have had a substantial impact on FI. Unfortunately, data from 2020 are not available to us. However, in order to be able to understand the impact of the pandemic on FI, a reliable baseline for comparison is needed. Thus, in the light of the COVID-19 pandemic, our results contribute by providing a baseline for comparing the experience of pre- and post-pandemic FS in the CEE in further studies. We realize however, that the issue of FI requires constant monitoring, which is a premise for further research in this area.

## 5. Conclusions

We analyzed profiles of FI using three categories: food security, mild FI, moderate or severe FI. We found that FI rates decreased with increasing severity. Our study revealed the presence of the FI problem in CEE countries and its various forms, which shows that

the topic is timely and important in the context of achieving sustainable development. We found differences in terms of profiles of FI in analyzed countries, which shows that it is worth analyzing the situation of FI in a given country separately. The identification of FI profiles can help in achieving the sustainable development goals at the country level. We found that the presence of FI translates into the assessment of satisfaction with the life of the population, which is in line with the hierarchy of human needs and confirms the importance of activities aimed at ensuring FS in the politics of each country. We examined the influence of socioeconomic and demographic correlates using multinomial logistic regression models, with food security as the reference category. Our findings reveal that there are significant distinctions in the relationships between individual characteristics and FI status. Our study explores the individual's FI profiles and provides evidence on the dependence of socioeconomic and demographic characteristics. Responding to the question of who are the food insecure in CEE countries, it was found that FI status was related to education, gender, age, household composition and income. Apart from typical correlates of FI, a remarkable negative effect of social capital on FI was found.

Our results show that the same socioeconomic and demographic discriminants determine the different probability of the occurrence of FI and its different forms in individual countries. Identifying groups particularly vulnerable to FI may allow appropriate targeting of instruments counteracting FI and adapt them to people with different demographic, social and economic characteristics. Awareness of the factors that are associated with FI should help to target those individuals most at the risk of FI, as well as focus policy recommendations.

**Author Contributions:** Conceptualization, H.D.; J.M.-R., A.W.-W.; methodology, H.D.; formal analysis, H.D.; data curation, H.D.; writing—original draft preparation, H.D.; J.M.-R., A.W.-W.; writing—review and editing, H.D.; J.M.-R., A.W.-W. All authors have read and agreed to the published version of the manuscript.

**Funding:** This research received no external funding. The article fee was financed by the Polish Ministry of Science and Higher Education within funds of the Institute of Economics and Finance and the Institute of Human Nutrition Sciences, Warsaw University of Life Sciences (WULS), for scientific research.

**Institutional Review Board Statement:** Not applicable.

**Informed Consent Statement:** Not applicable.

**Data Availability Statement:** Not applicable.

**Acknowledgments:** We thank FAO-VoH for the data license. Especially, the authors are grateful to Meghan Miller from the FAO Food Security and Nutrition Statistics Team for her able assistance and kind support in the FIES data acquisition process.

**Conflicts of Interest:** The authors declare no conflict of interest.

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
