# Peer review of "Profiles of Food Insecurity: Similarities and Differences across Selected CEE Countries"

_energies, doi:10.3390/en14165070_

Round 1

Reviewer 1 Report

This study of food insecurity in Lithuania, Poland and Slovakia covers and important topic in an understudied population.  The current version of the manuscript needs a fair amount of work to make it publishable.  I list my comments/concerns below:

  1. CEE should be spelled out the first time it is used (in the abstract)
  2. On page 4, Table 1 - Did you mean "rate" of change here, rather than "range of change?"  Also, it would be useful to put the average GDP per capita value on the table (not just the change)
  3. Better rationale is needed for the use of multinomial regression rather than ordinal regression.  If the authors are worried about failing the proportional odds assumption, there are other methods such as generalized ordered logit to get around this problem.  In regards to their point 2, ordinal models can do this as well.  If there are no other reasons to chose multinomial regression over ordinal regression, the authors should at least provide the ordinal models in an appendix as part of a sensitivity analysis.
  4. Figure 3 is confusing. I would either revise it or just describe the findings in the text.
  5. I think a better job of explaining to the reader why comparing Lithuania, Poland an Slovakia are interesting comparison cases, and what you get out of a comparison, rather than just reporting the country findings separately or all together.  To me, it currently reads as if they are being compared because the authors have the data.  There needs to be a better conceptual/theoretical reason for comparing these countries.
  6. The authors should report the standard errors for the results in Table 4
  7. Given how central inequality is in the food insecurity literature, some measure of income/economic inequality should be included in the models.
  8. pages 10-11 - the section that is entirely bullet points needs to be changed into actual text in paragraph form.  This is not appropriate for an academic article as it is currently presented.
  9. I don't think the ideal types that were constructed in Table 5 are useful.  In my opinion this doesn't add anything to the overall argument

Reviewer 2 Report

This is a rigorously done paper on an important topic . The authors should however provide a better incorporation in the literature on food (in)security and its drivers as well as the methods used there also considering the role of FI in the agecon sector. E.g. the following studies should be included: D'Souza (2020) who also apply a multinomial logit to assess FI and non-farm work. Other examples are Buhler et al (2018); 

Also regarding methods applied there should be a stronger motivation for the approach applied. E.g. why is the multinomial logit the adequate approach here compared to e.g. the multinomial endogeneous swtching model model used in Zhang et al (2021).

The named studies should be added

References:

Buhler, D., Harteja, R., & Grote, U. (2018). Matching food security and malnutrition indicators: Evidence from Southeast Asia. Agricultural Economics, 49(4), 481–495.

Campi et al. (2021) Specialization in food production affects global food security and food systems sustainability. World Development 141

D’Souza, A., Mishra, A.K. and Hirsch, S. (2020): Enhancing Food Security through Diet Quality: The Role of Nonfarm Work in Rural India. Agricultural Economics 51(1): 95-110.

Hirvonen et al (2021). Food Consumption and Food Security during the COVID-19 Pandemic in Addis Ababa. American Journal of Agricultural economics

Zezza, A., Calogero, C., Davis, B., & Winters, P. (2011). Assessing the impact of migration on food and nutrition security. Food Policy, 36(1), 1–6.

Zhang, J., Mishra, A.K., Hirsch, S (2021): Market-Oriented Agriculture and Farm Performance: Evidence from Rural China. Food Policy 100: 102023

Round 2

Reviewer 1 Report

The authors have done a solid job of responding to the comments.  I think the manuscript is publishable now.